



# Eddy kinetic energy and baroclinic and barotropic energy conversion rates along the Atlantic Water boundary current north of Svalbard

Kjersti Kalhagen[1,2], Ilker Fer[2,3,1], Till M. Baumann[4], Jon Albretsen[5], and Lukas Frank[1,2]

[1]Department of Arctic Geophysics, University Centre in Svalbard (UNIS), Longyearbyen, Svalbard, Norway
[2]Geophysical Institute, University of Bergen, Bergen, Norway
[3]Bjerknes Centre for Climate Research, Bergen, Norway
[4]Institute of Marine Research, Bergen, Norway
[5]Institute of Marine Research, Flødevigen Research Station, Aust-Agder, Norway

**Correspondence:** Kjersti Kalhagen (kjerstik@unis.no)

**Abstract.** On the continental slope north of Svalbard, the boundary current carrying Atlantic Water (AW) loses heat as it flows eastward. This cooling cannot be fully attributed to atmospheric heat loss or turbulent mixing. Lateral exchange, potentially linked to mesoscale activity, has previously been proposed as a contributing factor, based on limited observations of eddies. Here, we analyse a year-long dataset of hydrography and velocity observations from two mooring arrays, supplemented by

output from an eddy-resolving ocean model, to quantify the seasonal variability of eddy kinetic energy (EKE) and baroclinic and barotropic energy conversion rates over time-scales from days to months. Both EKE and conversion rates peak in autumn and winter, coinciding with the strongest boundary current and the warmest AW. Local EKE variability, however, is only weakly associated with conversion rates, suggesting advection from upstream generation sites or unresolved variability from limited measurements. Conversion is mainly baroclinic, indicating boundary current instability and providing conditions favourable

for offshore propagation of warm-core eddies. The model underscores the need for adequate spatial and temporal averaging and reveals substantial conversion rates on the offshore, deeper side of the boundary current, with comparable contributions from baroclinic and barotropic processes. Resulting mesoscale activity likely enhances lateral stirring and heat loss from the boundary current, particularly in winter and spring, contributing to the along-stream cooling of AW.

## 1 Introduction

Atlantic Water (AW) is the primary oceanic source of heat for the Arctic Ocean and plays a central role in shaping its changing physical environment (e.g., Carmack et al., 2015) and ecosystems (Ardyna and Arrigo, 2020). The largest volume of warm AW enters the Arctic Ocean with the West Spitsbergen Current (WSC), which flows through Fram Strait – the main deep gateway connecting the Arctic Ocean to the global oceans (Aagaard et al., 1987; Beszczynska-Möller et al., 2012). The AW flow splits into several poleward and recirculating branches as it approaches the Yermak Plateau (Fig. 1), and the inflow eventually

merges downstream and contributes to the Arctic Circumpolar Boundary Current (Rudels et al., 1999). The region north of Svalbard has been shown to have far-reaching signatures in the Arctic Ocean (Polyakov et al., 2017), such as recent changes



reported hundreds of kilometres downstream in the Eurasian Basin, including warming of the AW core (Richards et al., 2022), weakening of the halocline and shoaling of the AW (Polyakov et al., 2020b, a), thus affecting the exchange with sea ice and the surface layer.

The continental slope north of Svalbard is an important area for the modification of AW in the boundary current. The drivers of water mass transformation include air-sea-ice interactions and mixing (Renner et al., 2018; Kolås et al., 2020; Pérez-Hernández et al., 2019; Koenig et al., 2022b), entrainment of surface waters (Richards et al., 2022) and shelf waters (Schauer et al., 1997), and mesoscale variability and eddies (Koenig et al., 2022b; Crews et al., 2018; Wekerle et al., 2020; Athanase et al., 2020; Pérez-Hernández et al., 2017). We define the mesoscale as low Rossby number flows ($\mathrm{Ro} = U/fL < 1$, where $U$ and $L$

are the characteristic horizontal velocity and length-scales and $f$ is the Coriolis frequency) that occur on time-scales of days to months and length-scales of 10 km to 100 km. In particular, mesoscale eddies facilitate slope-basin exchange and lateral heat loss (Våge et al., 2016; Pérez-Hernández et al., 2017; Renner et al., 2018). In this study, we investigate the mesoscale eddy variability and energy conversion rates that transfer energy from the mean flow and stratification into eddy energy, using moored observations and a high-resolution numerical ocean model north of Svalbard.

Mesoscale eddies can form through barotropic and baroclinic instabilities of the flow (Cushman-Roisin and Beckers, 2011). Barotropic instability may arise from horizontal velocity shear, drawing kinetic energy from the mean flow and converting it into eddy kinetic energy (EKE) (e.g., Cushman-Roisin and Beckers, 2011; Teigen et al., 2010). For a boundary current over a steep slope, the shape of the current determines whether barotropic instability can form. Steep slopes stabilise the current (von Appen et al., 2016), requiring the current to be narrow and fast to become unstable (Håvik et al., 2017). Baroclinic instability,

in contrast, may form in regions with a horizontal density gradient, hence strong thermal-wind shear, and converts available potential energy into eddy energy, which enforces flattening of the pycnocline. Bottom slopes also impact (suppress) baroclinic instability (Isachsen et al., 2024).

In the Arctic Ocean, eddies play an important role in variability, transport of inflow waters, and ice–ocean interaction. High mesoscale variability is observed over continental slopes, in Fram Strait, and in the Arctic Circumpolar Boundary Current (von

Appen et al., 2022; Wang et al., 2020). These regions, together with the Barents Sea, have relatively higher EKE and baroclinic conversion compared to the rest of the Arctic Ocean (Li et al., 2024). Eddies that shed from unstable fronts along the boundary of the Arctic Ocean transport inflow waters across the basin (Carmack et al., 2015). Eddies also mediate ice–ocean interaction. Under-ice cyclonic eddies can bring AW upward into the surface layer and towards the ice, increasing the heat fluxes and enhancing melting (Müller et al., 2024). Cyclonic eddies may trap and advect sea ice into warmer waters (Manucharyan and

Thompson, 2017). Sea ice has been shown to dampen near-surface eddies (Meneghello et al., 2020). The ongoing decline of Arctic sea ice may reduce this effect, causing an increase in near-surface eddy activity and EKE (Armitage et al., 2020; Manucharyan and Thompson, 2022; Müller et al., 2024; Li et al., 2024).

In our study region, north of Svalbard, eddies have been registered both in situ and in models. Scarce observations show evidence for anticyclonic eddies carrying warm anomalies offshore from the shelf break (Våge et al., 2016). These eddies

laterally stir and redistribute heat, and thereby contribute to the heat loss from the boundary current (Renner et al., 2018; Koenig et al., 2022b). Estimates of along-path heat loss of AW revealed that turbulent heat fluxes and heat loss to the atmosphere alone





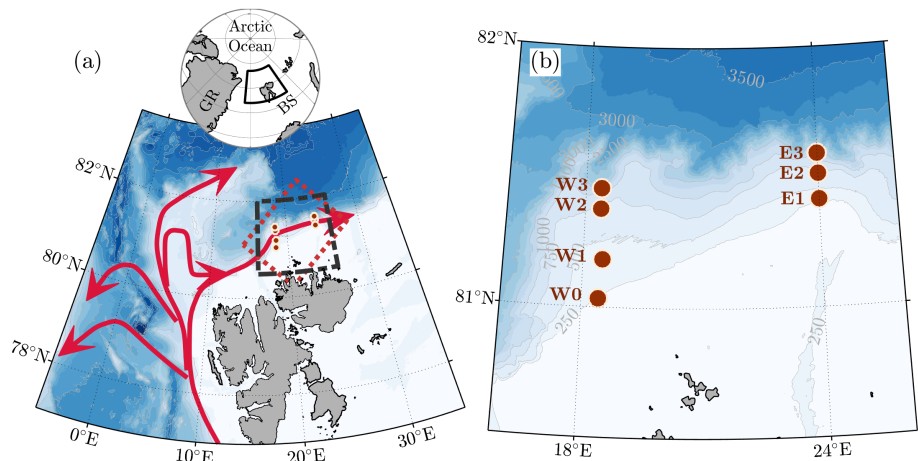

**Figure 1.** a) Bathymetry (shading and contours every 500 m; IBCAO version 4, Jakobsson et al. (2020)) and circulation of AW (red arrows) in Fram Strait and north of Svalbard. The black box outlines the map in (b), the red box shows the model subdomain. The inset in (a) shows the location in the Arctic Ocean (BS: Barents Sea, GR: Greenland). b) The study region showing the bathymetry and mooring positions.

could not account for the observed cooling, suggesting a substantial contribution through lateral heat loss (Koenig et al., 2022b). High-resolution model results support this view, showing anticyclonic eddies carrying AW spawning from the boundary current (Crews et al., 2018), propagating offshore towards the deeper basin (Wekerle et al., 2020), facilitating slope–basin exchange

and lateral transport of AW. However, the seasonal magnitude and variability of the associated energy conversion rates have not been previously reported from observations in this area.

In the Nansen LEGACY project, the slope north of Svalbard was chosen for targeted studies, including mooring arrays to quantify the properties of the boundary current, specifically its volume and heat transport, along-slope cooling (Koenig et al., 2022b), and the mesoscale variability (this study). These detailed observations are analysed here to quantify energy conversion

rates and are supplemented using outputs from an eddy-resolving ocean model. The combined results provide insight into where and how the boundary current flow energises variability at mesoscale eddy scales that could contribute to the lateral heat loss from the boundary current.

## 2 Data and Methods

### 2.1 Moorings

Ocean temperature, salinity, and horizontal current data obtained from two mooring arrays across the continental slope north of Svalbard are used in this study (Fig. 1). Measurements were collected for approximately one year from September 2018. The western array (18° E) was recovered in September 2019, and the eastern array (24° E) was recovered in November 2019. The data are available from Fer et al. (2022) and have been presented in Koenig et al. (2022b). Each mooring array consisted



**Table 1.** Mooring positions, total depth, and temporal coverage of the deeper mooring pairs used in this study.

| Mooring | Latitude | Longitude | Depth (m) | Deployed | Recovered |
|---------|----------|-----------|-----------|----------|-----------|
| W2 | N 81°22.686′ | E 18°23.789′ | 727 | 15.09.2018 1420 | 21.09.2018 0800 |
| W3 | N 81°27.356′ | E 18°23.730′ | 1202 | 20.09.2018 1810 | 21.09.2018 1330 |
| E2 | N 81°30.813′ | E 23°59.853′ | 706 | 16.09.2018 1120 | 23.11.2019 1905 |
| E3 | N 81°35.453′ | E 23°59.982′ | 1222 | 16.09.2018 1445 | 23.11.2019 2150 |

of one upper slope mooring (W1 and E1) near the $300\ \mathrm{m} - 400\ \mathrm{m}$ isobath, one middle slope mooring (W2 and E2) near the

700 m isobath, and one lower slope mooring (W3 and E3) near the 1200 m isobath, providing a good coverage of the boundary current (Fig. 1b). In addition, a mooring was deployed onshore of the western array (W0). In this study, we used data from the deeper pair of moorings, W2–W3 and E2–E3 (Table 1).

Mooring instrumentation coverage and setup, and details of data processing can be found in Koenig et al. (2022a). Temperature, salinity and horizontal current records were checked for inter-consistency and compared and corrected against ship-based

profiles taken at mooring deployment/recovery. All time series were hourly averaged (the sampling interval of instruments varied between 5 min and 1 hour) and linearly interpolated onto the same hourly time vector. At each hourly time step, instrument depths were obtained by linear interpolation between instruments equipped with a pressure sensor, to account for mooring knockdowns. The data were finally gridded onto a regular $(t, z)$ grid with 1 h temporal and 10 m vertical resolution.

## 2.2    Regional Ocean Model

In order to better evaluate and interpret the analysis based on the mooring observations, we used the output from a high-resolution ocean model applying ROMS (Regional Ocean Modeling System; Shchepetkin and McWilliams (2005)) in a domain covering the continental shelf and slope north of Svalbard (Frank et al., 2025). The model used the same configuration as the Norkyst model system documented in Asplin et al. (2020) but was set up with a horizontal resolution of 500 m and 35 vertical topography-following levels. The model configuration also included a sea ice module (Budgell, 2005). The operational

ocean forecasting model Barents2.5 (Röhrs et al., 2023) provided boundary conditions including tidal forcing, and the operational weather forecasting model AROME-Arctic (Müller et al., 2017) provided atmospheric input. The model simulation was initialised in April 2019, and we used two-year daily output from October 2019 to align seasonally with the mooring observations. Hourly saved fields were daily averaged, and the daily fields were then detided for the fortnightly and monthly tidal constituents using UTide (Codiga, 2011).

## 2.3    Methods for energy analysis

### 2.3.1    Wavelet analysis

In order to analyse the time-variability of energetic frequency bands, we used wavelet transforms with generalised Morse wavelets following Lilly and Olhede (2010, 2012) with the parameters $\gamma = 3$ for symmetric wavelets (Lilly and Olhede, 2009)





and $\beta = 9$ for a reasonable time and frequency resolution over the relevant time-scales studied here. We denote the wavelet
transform of a time series $x$ as $W_x$ and its complex conjugate as $W_x{}^*$.

### 2.3.2   Eddy kinetic energy and energy conversion rates

The calculations of eddy energetics and conversion rates require definition of fluctuations, background conditions, and aver-
aging. We use primed values (e.g., $u', \rho'$) to denote fluctuations from the mean, and an overbar to indicate the background
conditions and time averaging (e.g., $\bar{u}, \bar{\rho}$). For the moorings, fluctuations were obtained by band-pass filtering the hourly time
series using cut-off frequencies corresponding to 35 hours and two weeks, and the averaging was done over 30-day moving
windows. For the background conditions, we used a low-pass filter with a cut-off frequency corresponding to 30 days. From
the model output, primed values were obtained by Reynolds-decomposition, e.g., $u' = u - \bar{u}$, where $\bar{u}$ values are 30-day means
centred on the 15th of each month and $u'$ values are daily fluctuations from the mean.

Eddy kinetic energy density (EKE), the barotropic energy conversion (BT), and the baroclinic energy conversion (BC) rates
are defined as

$$\mathrm{EKE} = \frac{1}{2}\left(\overline{u'^2} + \overline{v'^2}\right), \tag{1}$$

$$\mathrm{BT} = -\rho_0\left(\overline{u'u'}\frac{\partial \bar{u}}{\partial x} + \overline{u'v'}\left(\frac{\partial \bar{u}}{\partial y} + \frac{\partial \bar{v}}{\partial x}\right) + \overline{v'v'}\frac{\partial \bar{v}}{\partial y}\right), \tag{2}$$

and

$$\mathrm{BC} = g\frac{\partial \bar{\rho}}{\partial z}^{-1}\left(\overline{u'\rho'}\frac{\partial \bar{\rho}}{\partial x} + \overline{v'\rho'}\frac{\partial \bar{\rho}}{\partial y}\right), \tag{3}$$

where $\rho_0 = 1027 \ \mathrm{kg\,m^{-3}}$ is a reference density, $\overline{u'u'}$, $\overline{u'v'}$, and $\overline{v'v'}$ are eddy momentum fluxes, $g$ is the gravitational acceler-
ation, $\overline{u'\rho'}$, $\overline{v'\rho'}$ are eddy density fluxes, $\partial/\partial y$ and $\partial/\partial x$ denote lateral gradients, and $\partial\bar{\rho}/\partial z$ is the mean vertical stratification.

In the model subdomain, EKE, BT, and BC were calculated on the model grid. Then, $x$ and $y$ denote the right-handed
orthogonal coordinates in the native model grid, and $u$ and $v$ the velocity components in these directions. For the moorings,
$x$ and $y$ denote the local along- and across-slope directions, and $u$ and $v$ the along- and across-slope current components. The
coordinate system was rotated with $20°$ at the western and $-5°$ at the eastern array, respectively.

### 2.3.3   Simplified calculations for conversion rates

Conversion rate estimates from the mooring records were made using simplified forms of Equations 2 and 3, with the common
assumptions that the across-isobath gradients dominate and that there is no along-isobath variability:

$$\mathrm{BT} = -\rho_0\,\overline{u'v'}\frac{\partial \bar{u}}{\partial y}, \tag{4}$$

$$\mathrm{BC} = g\,\overline{v'\rho'}\frac{\mathrm{d}z}{\mathrm{d}y}, \quad \frac{\mathrm{d}z}{\mathrm{d}y} = \frac{\partial \bar{\rho}}{\partial y}\bigg/\frac{\partial \bar{\rho}}{\partial z}. \tag{5}$$





In BT (Eq. 4), the lateral gradient was obtained by first differencing of $u$ that was layer-averaged between 300 m and 700 m (280 m and 680 m at the eastern array), and the eddy momentum flux was calculated at each mooring and then averaged to be representative of the flux between the moorings. For layer averages, standard error is calculated as $\mathrm{SE}(t) = \sigma(t)/\sqrt{n(t)}$, where $n(t)$ is the number of estimations at time $t$, and $\sigma(t)$ is the standard deviation of the $n$ estimations. In BC (Eq. 5), $\mathrm{d}z/\mathrm{d}y$ is the mean isopycnal slope, estimated by the quotient of the across-slope gradient of the mean density $\partial\bar{\rho}/\partial y$ and the mean stratification $\partial\bar{\rho}/\partial z$. The potential density $\rho$ was calculated from gridded $T$ and $S$ fields, at selected depths near the target depths of the sensors, to minimise error from linear interpolation across the halocline.

We conducted additional calculations of BT and BC from the model output using the simplified equations (Eqs. 4 and 5) applied to the time series of velocity and density extracted at locations representative of the moorings ("virtual moorings") as well as 70 km and 50 km transects ("segments") with 101 and 71 grid points across the slope, covering the virtual moorings. Virtual mooring calculations mimic the data and the method used for the in situ moorings, which we compare to the full volume-averaged or segment-averaged conversion rates in order to assess their limitations.

### 2.3.4 Calculations along the boundary current pathway

We analysed the EKE and energy conversion rates obtained from the model to describe the temporal and spatial variability along the pathway of the boundary current. Upon inspection of the model outputs, we identified the 1400 m isobath to be representative of the energetic part of the boundary current on the upper slope. The structure along the isobath is obtained by smoothing the monthly fields using a 2D Gaussian filter with a 20 km length-scale and interpolating the results onto the isobath. The smoothing length-scale ensures that we capture the boundary current structure between, typically, 800 m and 2000 m isobaths.

As an indication for conditions that might allow barotropic instability along the boundary current pathway, we calculate the cross-isobath gradient of potential vorticity, $q_y = \beta_{\mathrm{topo}} - \partial^2\bar{u}/\partial y^2$, where $\beta_{\mathrm{topo}} = -(f/H)\,\partial H/\partial y$ is the topographic $\beta$, $H$ is the water depth, $f$ is the Coriolis parameter, $u$ is the along-stream current velocity component, and $y$ is the across-slope direction. A necessary, but not sufficient, condition for barotropic instability is that $q_y$, must change sign within the domain (Vallis and Maltrud, 1993). Along the 1400 km isobath, we calculated $q_y$ on 20 km lines oriented perpendicular to and centred on the isobath. To approximately visualise this instability condition, we show the product of the minimum and maximum values of $q_y$ calculated along each line, as a change in sign of $q_y$ would result in negative values.

## 3 Results

### 3.1 Seasonal and mesoscale variability of the boundary current

Mean structure and seasonal variability of the boundary current observed in the mooring arrays have been reported in Koenig et al. (2022b). Here, we summarise the mean hydrography and current structure relevant to our analysis (Fig. 2) using the three-monthly averages from Koenig et al. (2022b) but grouped into September–February and March–August periods.





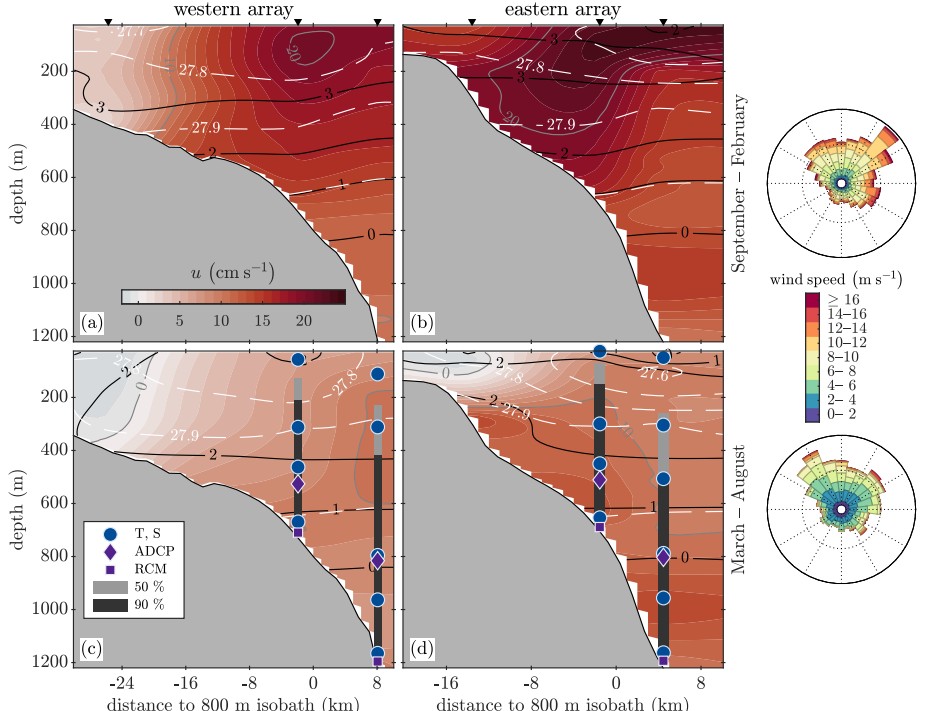

**Figure 2.** Average along-slope current velocity $u$ (colours) on (a, c) the western and (b, d) eastern mooring arrays during (a, b) autumn and winter (September–February) and (c, d) spring and summer (March–August). Gray contours show velocity every $10 \ \mathrm{cm \ s^{-1}}$, white dashed isopycnals every $0.1 \ \mathrm{kg \ m^{-3}}$, and black isotherms every $1 \ ^\circ\mathrm{C}$. Triangles at the top mark the mooring locations (except W0). The two deeper mooring pairs used in this study are illustrated in the lower panels: vertical bars show the current profiling coverage (light gray: 50 % and dark gray: 90 % of the time); circles indicate CTD sensors, diamonds ADCPs, and squares RCMs. Wind roses (right) show the relative frequency of wind direction and speed during (a, b) September–February and (c, d) March–August in the region $18^\circ$ E–$24.5^\circ$ E, $81.2^\circ$ N–$81.7^\circ$ N, extracted from the Copernicus C3S Arctic Regional Reanalysis (CARRA, Schyberg et al., 2020).

The boundary current was on average stronger from September to February and weaker from March to August (Fig. 2). In September–February, the velocity core was at around 100 m depth over the 800 m isobath, carrying AW warmer than $3 \ ^\circ\mathrm{C}$ at a speed exceeding $20 \ \mathrm{cm \ s^{-1}}$ (Fig. 2a–b). In March–August, the velocity core was weaker ($10 \ \mathrm{cm \ s^{-1}}$ to $20 \ \mathrm{cm \ s^{-1}}$), deeper, and relatively diffuse in both vertical and lateral directions (Fig. 2c–d). Water with the highest temperatures, exceeding $2.5 \ ^\circ\mathrm{C}$, was at 200 m depth seawards of the 500 m isobath, but did not coincide with the deeper velocity core.

The mesoscale activity was elevated in autumn and winter (Fig. 3), the same period when the boundary current was strongest and carried the warmest water (Fig. 2a–b). The strongest velocity variability occurred on time-scales between 2 and 6 days and, like the elevated mesoscale activity, was observed in autumn and winter (Fig. 3a, c). From October 2018 to January 2019, the offshore velocity component $v$ and the temperature $T$ co-varied on a 3–10-day time-scale (Fig. 3d). At that time, the current core was located near the 800 m isobath, close to W2, and carried AW (Fig. 2a).



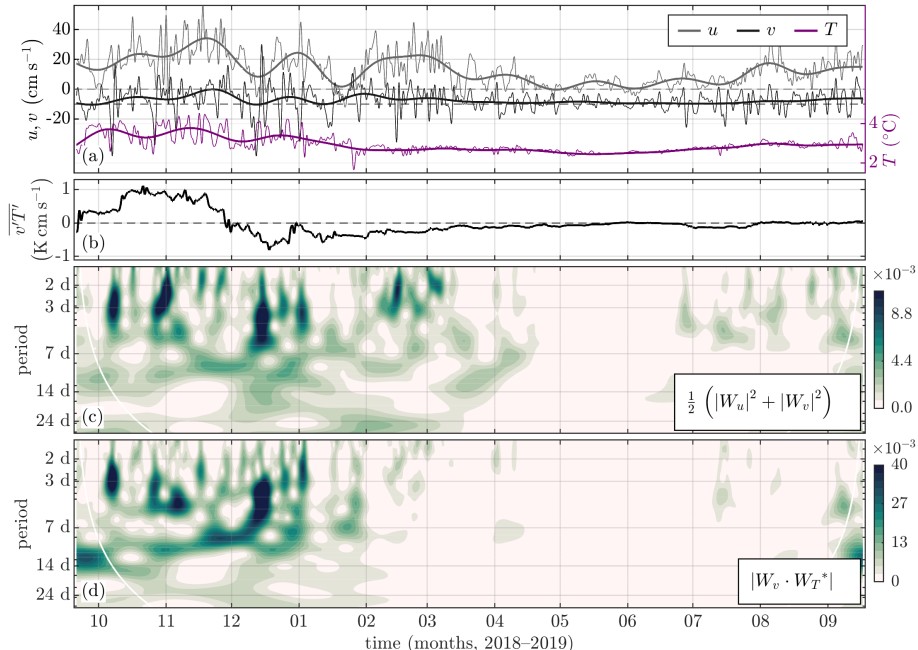

**Figure 3.** Time series of a) along-slope and across-slope current velocity $u$, $v$ (left axis; $v$ is offset by $-10\,\mathrm{cm\,s^{-1}}$), and temperature $T$ (right axis, purple) measured at 250 m depth at the western mooring W2 (a level in the core which consistently has both temperature and velocity measurements), low-passed filtered at 35 hours (thin curves) and 30 days (thick curves). Corresponding b) offshore eddy temperature flux $\overline{v'T'}$ (positive for warm anomalies towards deeper water and cold anomalies towards shallower water), c) wavelet power spectrum, $\frac{1}{2}\left(|W_u|^2 + |W_v|^2\right)$, and d) wavelet cross-spectrum $|W_v \times W_T{}^*|$ between $v$ and $T$. The cone of influence (solid white curve) in (c, d) indicates the areas affected by boundary effects.

The offshore eddy temperature flux, $\overline{v'T'}$, at 250 m depth was positive in the autumn of 2018, peaking in mid-November, and became negative in December 2018 (Fig. 3b), consistent with mesoscale activity driving cross-isobath heat transport. Averaged over September–November 2018, $\overline{v'T'}$ at W2 was positive at depths 150 m to 350 m, spanning the temperature core. At W3,

170 the eddy temperature flux in the upper water column could not be calculated in autumn due to mooring knockdown, but was positive at and below 400 m depth. The eddy temperature flux divergence between W3 and W2 was over four times higher in autumn and winter (September–February) than in spring and summer (March–August), suggesting stronger lateral heat loss from the boundary current in autumn and winter.

While the relatively higher eddy temperature flux and -divergence in autumn and winter indicate increased mesoscale activ-

175 ity, we cannot directly quantify net lateral heat transport because the rotational and the divergent components of the flux could not be separated (Marshall and Shutts, 1981; Guo et al., 2014).





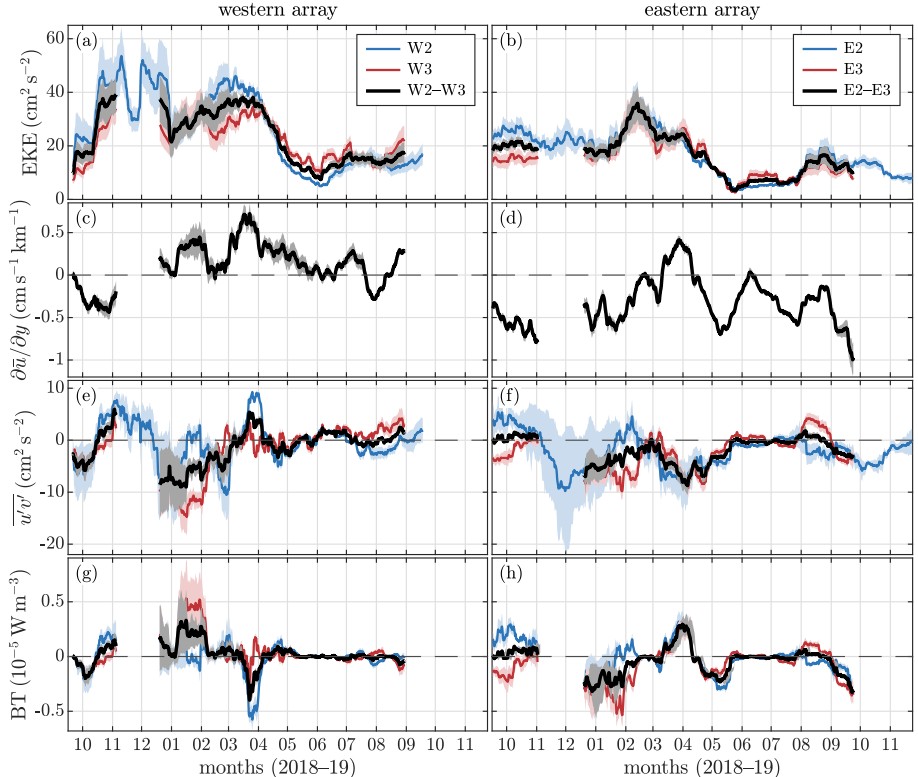

**Figure 4.** Time series of (a, b) EKE, (c, d) lateral shear of the mean along-slope current, (e, f) eddy momentum flux, and (g, h) BT at the (left) western and (right) eastern arrays. Blue curves are values at W2 and E2, red curves at W3 and E3, and black curves for the average between the pair of moorings. Shading shows standard error calculated over estimations at every hundred meters between 300 m and 700 m depth.

## 3.2 Energetics and conversion rates from mooring data

EKE was higher in autumn and winter than in spring and summer at both mooring arrays (Fig. 4a, b), and it was surface intensified with values at 300 m depth about four times higher than at 700 m (not shown). In contrast, BT remained low year-round at both sites (Fig. 4g, h), negligible in summer and varying within $\pm 0.5 \times 10^{-5}$ W m$^{-3}$ otherwise. The low BT resulted from a combination of averaging oppositely signed momentum fluxes (Fig. 4e, f) and occasionally low velocity shears (Fig. 4c, d).

At the western array, the autumn- and wintertime EKE reached nearly $40\,\mathrm{cm}^2\,\mathrm{s}^{-2}$, while the summertime EKE dropped below $20\,\mathrm{cm}^2\,\mathrm{s}^{-2}$ (Fig. 4a). BT was mostly low except for two events: In January 2019, BT was positive (Fig. 4g) when the values in the upper water column reached $1 \times 10^{-5}$ W m$^{-3}$ due to negative momentum flux and positive shear. These decreased with depth, reducing the layer-average BT. In late March 2019, simultaneous strong positive velocity shear and positive momentum flux at W2 resulted in a similarly large but negative and shorter-lasting BT (Fig. 4g, c, e).



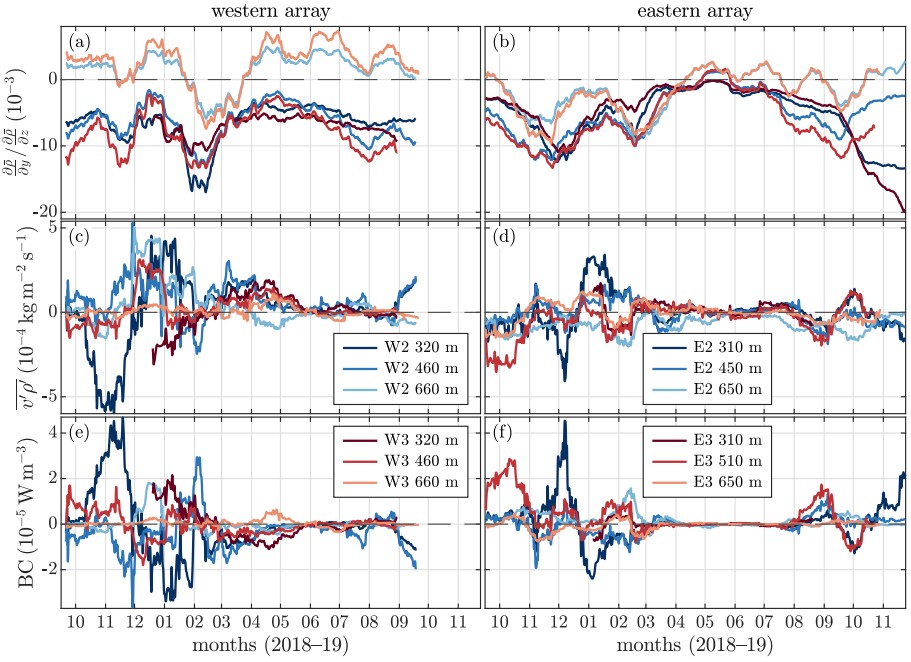

**Figure 5.** Time series of (a, b) isopycnal slope, (c, d) eddy density flux, and (e, f) BC at the (left) western and (right) eastern arrays. A positive isopycnal slope occurs when density decreases with offshore distance.

The eastern array had generally lower EKE than the western, but with a similar seasonal contrast: autumn–winter values reached or exceeded $20\,\mathrm{cm^2\,s^{-2}}$, peaking at $35\,\mathrm{cm^2\,s^{-2}}$ in February 2019 (Fig. 4b). BT in the eastern array was quite similar to BT in the west, generally low but with two events. The eastern conversion events appeared with opposite signs as the ones in the western array. BT was negative in January 2019 due to modest momentum fluxes and shear, while from late March to early April 2019, BT was positive due to a period of negative momentum fluxes and an offshore drift of the boundary current (Fig. 4h, d, f). When EKE peaked in February, BT was negligible due to near-zero shear and fluxes.

BC was generally largest in autumn and winter at both mooring arrays, ranging within $-3 \times 10^{-5}\,\mathrm{W\,m^{-3}}$ and $4 \times 10^{-5}$ $\mathrm{W\,m^{-3}}$, decaying to negligible values through spring and summer (Fig. 5e, f). BC values were typically largest in the upper water column, due to higher fluctuations in density and cross-slope velocity, and decreased with depth (Fig. 5c, d). Notably, BC estimates differed substantially between depths and moorings, especially during autumn and winter (Fig. 5e, f), mainly due to differences in eddy density fluxes despite being measured only 100 m apart vertically, e.g. in October and November 2018 (Fig. 5c, d).

At the western array, the most notable BC event occurred at 320 m at W2 from mid-October through November 2018. BC increased from negligible values to $4 \times 10^{-5}\,\mathrm{W\,m^{-3}}$ and remained high for nearly a month. This event was associated with a strong mean current, high EKE (Fig. 4a), a modest and steady negative isopycnal slope, and an elevated eddy density flux due to an increased variability and co-variability of $v$ and $\rho$ of periodicity 2–10 days (similar to Fig. 3d). During and following





this event, EKE at W2 was high and peaked at $88\,\mathrm{cm}^2\,\mathrm{s}^{-2}$ at 300 m depth. However, such coincident increases in EKE and BC were not observed frequently. A similar but shorter-lasting event was observed at 310 m depth at E2 from mid-November to early December 2018 (Fig. 5f). Here, an eddy density flux of $-4 \times 10^{-4}\,\mathrm{kg}\,\mathrm{m}^{-2}\,\mathrm{s}^{-1}$ resulted in BC exceeding $4 \times 10^{-5}$ $\mathrm{W}\,\mathrm{m}^{-3}$. EKE at 280 m depth reached a local maximum of $55\,\mathrm{cm}^2\,\mathrm{s}^{-2}$ during this event (not shown).

In summary, the current strength, temperature variability, EKE, the eddy momentum and density fluxes, BC, and to some degree BT, were all generally stronger in autumn and winter compared to spring and summer. EKE, BT, and BC were generally surface intensified. While EKE was relatively enhanced during the months with higher fluxes and BC, there was no consistent or strong relationship between them.

### 3.3  Energetics and conversion rates from the ocean model data

We analyse the output from the high-resolution model to evaluate whether the conversion estimates using the simplified expressions for BT and BC from limited depth levels and locations of the mooring records are representative for a larger area. In order to assess this, we calculate EKE, BT, and BC using all terms of the conversion equations (Eqs. 2, 3) and averaging in the vertical between 100 m and 2000 m depth to obtain the spatial structure (Fig. 6). We next show the temporal evolution of the energetics averaged along the 1400 m isobath (Fig. 7). Additionally, we use the simplified equations (Eqs. 4, 5) applied to time series of velocity and density extracted at the mooring locations from the model ("virtual moorings"), using the same calculation method as for the moorings (Fig. 8).

#### 3.3.1  Mean spatial distribution

EKE values along the slope, averaged over 2 years, were typically $20 - 30\,\mathrm{cm}^2\,\mathrm{s}^{-2}$, with higher values, up to $45\,\mathrm{cm}^2\,\mathrm{s}^{-2}$, e.g. around and west of the western mooring array (Fig. 6a). The largest BT and BC magnitudes were located on the steep slope offshore of the 1000 m isobath (Fig. 6b, c). Areas where the two-year mean EKE, BT, and BC all exceeded high-percentile values were primarily located near between the 1000 m and 1800 m isobaths in the vicinity of the western mooring array. Such areas tended to be more localised for BT and BC but parts of larger maxima for EKE.

Along the slope, BT showed a pattern with positive values where isobaths veer leftward downstream, and negative values in areas where isobaths veer rightward (Fig. 6b). These zones were persistent in time (not shown). An example of two such oppositely signed zones is near the western mooring array where the values reached $\pm 7 \times 10^{-5}\,\mathrm{W}\,\mathrm{m}^{-3}$ in the two-year average. A particularly strong positive BT region was near the 1400 m isobath approximately 20 km downstream from the western array, where the average BT exceeded $20 \times 10^{-5}\,\mathrm{W}\,\mathrm{m}^{-3}$.

BC had comparable spatial variability and magnitude to BT (Fig. 6c). A hot-spot upstream of the western array averaged to $24 \times 10^{-5}\,\mathrm{W}\,\mathrm{m}^{-3}$, coincident with substantial EKE. The strong BT region 20 km downstream of the western array also had large BC occurring near the 1400 m isobaths.



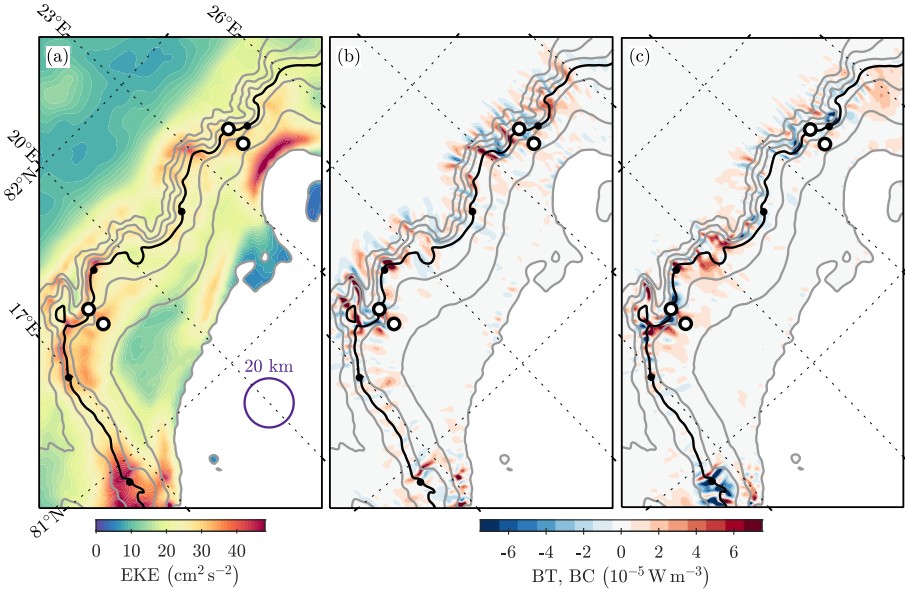

**Figure 6.** (a) EKE, (b) BT, and (c) BC in the model subdomain averaged over the two years from October 2019 to September 2021. The positions of the virtual moorings are marked with circles. Isobaths are drawn every 400 m between 200 m and 2600 m (gray), and the 1400 m isobath is highlighted in black. Black dots mark the distance along the 1400 m isobath every 50 km starting at 0 km with the first dot. The colour bar ranges are set to the 99.5 percentiles of the time-averaged fields. In (a), the circle at $20°$ E of diameter 20 km shows the length-scale of the spatial filtering applied in Fig. 7.

### 3.3.2 Structure along the boundary current pathway

Along the 1400 m isobath, EKE, BT, and BC varied both spatially and temporally (Fig. 7). Time-mean EKE values were elevated near the two mooring arrays and approximately 40 km upstream, but the highest mean EKE occurred at the southwestern boundary of the domain (Fig. 6a, Fig. 7a). Monthly averages over the span of the isobath ranged from below $20\,\mathrm{cm^2\,s^{-2}}$ to above $60\,\mathrm{cm^2\,s^{-2}}$ in month 14 (November).

BT averaged to $0.34 \times 10^{-5}$ W m$^{-3}$ along the 1400 m isobath over the full period. Its time-mean was mostly positive,
ranging from $-0.9 \times 10^{-5}$ W m$^{-3}$ to $2.2 \times 10^{-5}$ W m$^{-3}$, peaking about 20 km downstream of the western array, with a secondary maximum of $1.7 \times 10^{-5}$ W m$^{-3}$ located 15 km downstream of the eastern array. When averaged along the isobath, BT remained generally positive and reached its maximum value of $1.5 \times 10^{-5}$ W m$^{-3}$ in month 14, which coincided with the EKE maximum. No clear seasonality was observed.

The time-mean BC varied within $\pm 2 \times 10^{-5}$ W m$^{-3}$. The largest positive BC was approximately 10 km upstream and in a
wide zone 15 km to 45 km downstream of the western array. Near the eastern array, BC was negative on average. Averaged along the isobath, BC was typically within $\pm 0.5 \times 10^{-5}$ W m$^{-3}$, except for a couple of larger minima and a peak of $0.6 \times 10^{-5}$ W m$^{-3}$ in month 14, again coinciding with the EKE maximum.





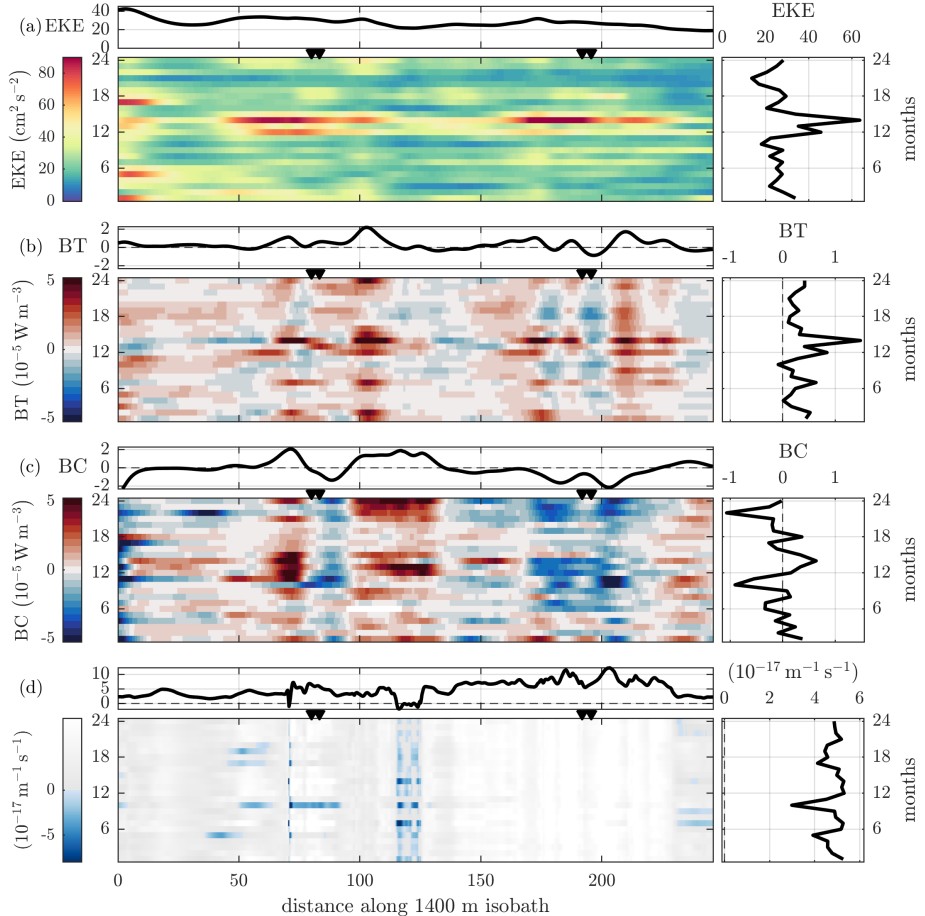

**Figure 7.** Spatio-temporal evolution along the 1400 m isobath of modelled a) EKE, b) BT, c) BC, and d) a measure of cross-isobath sign-change of the lateral PV gradient, $q_y$. Temporal averaging is 1 month and the spatial smoothing is 20 km across the isobath. (d) shows the product of the minimum and maximum of $q_y$ calculated across the isobath (negative values are indicative of a change in sign). Line plots in horizontal panels are time averages and those in vertical panels on the right are along-isobath averages. Triangles mark approximate locations of the virtual moorings. Distance-axis tick labels every 50 km correspond to the dots on the 1400 m isobath in Fig. 6.

Although there were occasional coincidences between peaks in EKE and conversion rates on the 1400 m isobath, the variability of EKE generally did not match the local conversion rates. This lack of a robust relation between EKE variability
and conversion rates mirrors the findings in the mooring observations and suggests that increasing EKE in a region does not necessarily indicate that strong local energy conversion is taking place.

The cross-isobath gradient of the effective potential vorticity, $q_y$, suggests some regions along the boundary current may experience barotropic instability. At each along-isobath point, the product of the minimum and maximum values of $q_y$ across the 1400 km isobath visualises whether $q_y$ changes sign (Fig. 7d) and is discussed further in Section 4.4. Some zones were
primarily near-zero or negative, e.g., a narrow zone 10 km upstream and a wider zone 40 km downstream of the western




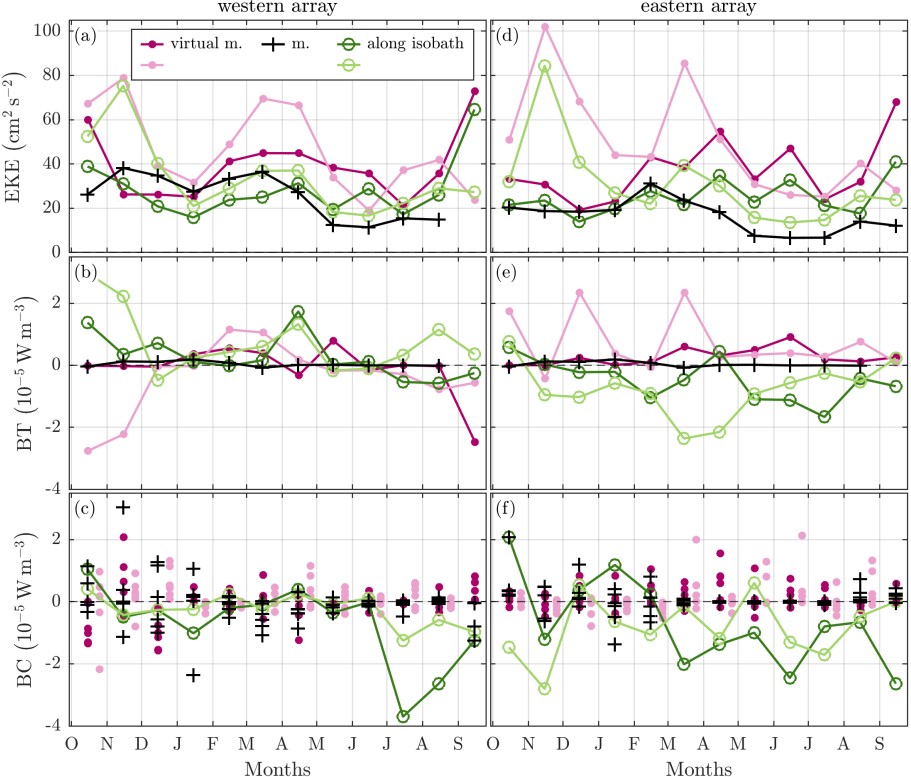

**Figure 8.** Time series of (a, d) EKE, (b, e) BT, and (c, f) BC from in situ moorings (black pluses), from the model using virtual moorings (purple dots), and the average along the 1400 m isobath near moorings (green open circles). The first model year is shown in dark, and the second year in pale colours. In (c, f), BC are shown for the same levels as the moorings (Fig. 5), and pale purple dots are offset by five days for clarity.

moorings, indicating that barotropic instability may occur locally with the current conditions in those two years. Other zones had both signs, indicating that topographic $\beta$ was low enough to be outweighed by the curvature of the current at the 1400 m isobath during some months. Some positive BT was found even where the necessary condition for barotropic instability was not met, indicating that eddy–mean flow interaction may occur in regions that are not locally unstable.

### 3.3.3 Variability at the virtual moorings

Virtual moorings reproduced, although slightly overestimated, the seasonal variability of EKE seen both in the observations and in the spatially averaged model estimates at each mooring site along the isobath (Fig. 8a, b). BT estimated from the virtual moorings was generally larger than that from the in situ moorings (Fig. 8b, e). BC estimates from virtual moorings showed variability similar to those from in situ moorings and depended similarly on sampling depth and the specific mooring pair used (Fig. 8c, f). We note that although the virtual and in situ moorings were placed at approximately the same geographical and





slope locations, they still might not be equivalently placed relative to the boundary current, and note also that the modelled and observed data sets cover different time periods.

Virtual moorings agreed fairly well with the simplified estimates of BT made on cross-slope segments (not shown). However, using the full BT equation and averaging over the full depth and across the region, the resulting spatial averages differed

substantially from the simplified virtual mooring and segment-based estimates, highlighting the effects of both the calculation method and the degree of averaging. Segment-based averages of BC were generally smaller than point-wise virtual mooring estimates (not shown), showing that spatial averaging tends to reduce variability. As with BT, spatially averaged BC values also diverged from virtual mooring results.

## 4  Discussion

### 4.1  Enhanced EKE and conversion in autumn and winter

We observed higher EKE during autumn and winter from the mooring observations (Fig. 4a–b). This agrees with Crews et al. (2018) who found higher EKE between October and March in a high-resolution model and with Koenig et al. (2022b) who found increased geostrophic surface EKE in the autumn and winter months. Upstream, the WSC is most energetic from December to May, with a peak in February (von Appen et al., 2016). In AW inflow regions, such as the Barents Sea Opening,

Fram Strait, and the western Nansen Basin, wintertime EKE can be relatively higher. This is attributed to the typically lower sea ice cover in these areas, which permits stronger wind forcing (von Appen et al., 2022). In addition, the lower sea ice cover in the inflow zones supports weaker stratification and higher instability (von Appen et al., 2022).

During the same periods of elevated EKE – autumn and winter – we also observed larger energy conversion, with BC as the dominant contribution (Figs. 4g–h, 5e–f). The important role of BC in autumn and winter seen in the mooring observations is

supported by the model, lending confidence on the simplified calculations from the moorings. Previous studies from upstream regions in Fram Strait and west of Spitsbergen have similarly shown that the boundary current is baroclinically unstable (Teigen et al., 2011; von Appen et al., 2016; Wekerle et al., 2020). The high wintertime EKE in the WSC has been linked to baroclinic instability due to weak stratification and a stronger vertical shear (von Appen et al., 2016). Similarly, in the Eurasian Basin, Müller et al. (2024) reported a spatial agreement between EKE and BC, suggesting that baroclinic instability supplies eddy

energy locally. Our finding that both EKE and BC are greater in autumn and winter indicates that baroclinic instability may similarly drive higher EKE north of Svalbard as well.

### 4.2  Comparison to other regions

Conversion rates have been estimated in several other studies using mooring observations and models. Across the pathway of the WSC in Fram Strait, conversion rates inferred from moored observations were highest near the shelf break during winter,

with a pronounced seasonal cycle, peaking in February and reaching its minimum in August (von Appen et al., 2016). Their largest BC value, $15 \times 10^{-5}$ W m$^{-3}$ at 75 m depth in February, is higher than our November maximum of $4 \times 10^{-5}$ W m$^{-3}$ at



320 m. At the same location, their BT in February reached $2.5 \times 10^{-5} \, \mathrm{W \, m^{-3}}$, which is about ten times higher than our layer-averaged maximum during January 2019 at the western array. However, in our data both BT and BC were larger at shallower depths between roughly 300 m and 700 m, so it is plausible that values at 75 m could be closer to those of von Appen et al.
300 (2016).

Further downstream of WSC in Fram Strait, Fer et al. (2023) estimated conversion rates from mooring records on the southern flank of the Yermak Plateau. Similar to our results, they observed that both BT and BC were highly variable in depth and time, with periods of positive and negative values.The conversion rates showed no strong correlation with EKE, consistent with our findings from both mooring and model data. Their observed EKE levels exceeded the local conversion rates, and could
be partly accounted for advection from an energetic region upstream of the moorings reported in von Appen et al. (2016).

Conversion rates in Fram Strait have also been estimated using two high-resolution models (Wekerle et al., 2020), resulting in values along the slope within $\pm 5 \times 10^{-5} \, \mathrm{m^3 \, s^{-3}}$ (equivalent to $\pm 5 \times 10^{-5} \, \mathrm{W \, m^{-3}}$ at 1000 m depth). BT had alternating sign along the slope, indicating a strong influence of bathymetry, consistent with our findings, a point we return to in Section 4.4.

Negative conversion rates, observed in our mooring records (lasting weeks) and in the model (lasting months in some
regions), have also been reported elsewhere (e.g., Spall et al., 2008; Håvik et al., 2017; Wekerle et al., 2020; Fer et al., 2023). Theoretically, negative BT implies eddy energy feeding the mean flow, while negative BC implies energy transfer that steepens isopycnals by lifting denser water onto the shelf. In our mooring data, a period of negative BC in January–February 2019 was followed by an increase in isopycnal slope (Fig. 5c, a). Due to uncertainties in the estimates from the moorings, we cannot determine whether this reflects a physical process or the observed isopycnal slope variability coincidentally aligned with the
estimated negative BC. However, negative conversion rates seen in both our measurements and in high-resolution ocean model studies support the interpretation that such processes may occur in the ocean, although further investigation is required to confirm this.

### 4.3 Mesoscale activity and heat loss from the boundary current

Understanding the heat loss of AW as it flows in the boundary current is key to regional climate and oceanography north of
Svalbard. Koenig et al. (2022b) estimated the seasonal evolution of along-stream heat loss of AW between the western and the eastern mooring arrays and showed that air–sea heat fluxes explain much of the AW cooling between 18°E and 24°E during autumn, but are insufficient to account for the cooling rates observed in winter, spring, and summer. This implies that additional mechanisms contribute to the along-stream heat loss of AW, such as lateral exchange proposed by Crews et al. (2018) and Kolås et al. (2020).

Several observations from the slope north of Svalbard indicate the presence of such lateral processes: temperature variability exhibited an onshore–offshore mode with a 6–7 day periodicity that was correlated with across-slope velocity variability, and surface geostrophic EKE was elevated near the current core (Koenig et al., 2022b). Anticyclonic eddies are a likely agent for this exchange. Limited observations north of Svalbard showed anticyclones carrying warm anomalies offshore from the boundary current (Våge et al., 2016) and other eddy features linked to baroclinic instability (Pérez-Hernández et al., 2017),
which could contribute to lateral heat loss from the current.





Anticyclonic eddies carrying AW or warm anomalies have also been identified in high-resolution modelling studies both north of Svalbard (Crews et al., 2018) and upstream (Wekerle et al., 2020). Model results reported here also indicate substantial conversion rates, not only in the upper slope where moorings were placed but also farther offshore (Fig. 6). This offshore activity agrees with earlier observations of eddies over the deeper slope (Våge et al., 2016; Pérez-Hernández et al., 2017), as well as frequent offshore eddy propagation detected in a high-resolution model (Crews et al., 2018). Furthermore, from their mooring at the 800 m isobath, Renner et al. (2018) detected few eddies, suggesting that most warm-core eddies detach further offshore. Our observations and model results, together with previously published evidence, indicate mesoscale activity consistent with eddy-driven exchange contributing to AW heat loss from the boundary current.

Using the mooring arrays analysed here, Koenig et al. (2022b) reported that the along-stream heat loss of AW in autumn could be accounted for by the vertical heat loss to the atmosphere. During this period, the study area was ice free, and the AW core was at its warmest and closest to the surface. In winter and spring, however, when the AW core subducted and sea ice cover limited direct atmospheric cooling, $75\,\mathrm{W\,m^{-2}}$ out of $302\,\mathrm{W\,m^{-2}}$ along-stream cooling in winter, and $35\,\mathrm{W\,m^{-2}}$ out of $60\,\mathrm{W\,m^{-2}}$ in spring, was unaccounted for by heat loss to the atmosphere. Our analysis of the measured EKE and conversion rates suggests that the contribution of mesoscale activity can explain much of this missing heat loss from the boundary current in spring and winter (Fig. 4a, b).

During summer, however, when the surface gained heat from the atmosphere, Koenig et al. (2022b) estimated average along-stream cooling of AW comparable to winter, suggesting that all the heat loss in summer must be lateral or downward. EKE in this period was, however, at its lowest (Figs. 3c, 4a–b), and the calculated conversion rates were negligible (Figs. 4g–h, 5e–f). This implies that mesoscale eddies were unlikely to be the dominant agents for heat loss in summer, and other processes must play a role. A likely candidate for further lateral heat loss is topographic vorticity waves, which have previously been linked to increased heat loss from the WSC (Nilsen et al., 2006). An ongoing study (unpublished) indicates that the topographic slope, together with the lower stratification and slower current observed in summer, could allow for vorticity waves with diurnal periods, which are energized by the diurnal tide in the area near the western mooring array.

## 4.4 Mechanisms behind the energy conversion

The pattern of alternating signs of energy conversion rates along the continental slope (Fig. 6b) is consistent with findings from other regions such as Fram Strait (Wekerle et al., 2020) and the Lofoten escarpment (Fer, 2020). North of Svalbard, these time-persistent patterns were connected to the veering of the isobaths (Section 3.3.1), indicating the close connection to bathymetry noted by Wekerle et al. (2020). This connection is likely a manifestation of the necessary condition for barotropic instability, which depends on the variation in bathymetry through $\beta_{\mathrm{topo}}$.

Our quantification of the lateral sign change of the PV gradient (Fig. 6d) indicated some areas along the 1400 m isobath where the necessary condition for barotropic instability could be satisfied. However, the spatial and temporal patterns of BT are not directly explained by these areas (Fig. 6b): BT was positive in several regions – for example upstream of the eastern array – where the condition for instability was never met, and conversely, BT was low or negative in some areas where the condition was fulfilled. The latter can happen since meeting the necessary condition does not ensure instability will occur. Overall, the





connection between the potential for barotropic instability and the observed barotropic conversion was inconclusive. This can partly be explained by the mean current advecting instabilities or eddies from upstream, where they were triggered in areas satisfying the instability condition. Similar observations were made by e.g., Håvik et al. (2017).

For baroclinic instability to occur, the cross-stream gradient of the effective potential vorticity must change sign with depth (Charney and Stern, 1962; Spall and Pedlosky, 2008). With only two moorings at each array, we cannot reliably determine
whether this condition is met. However, small values of the geostrophic Richardson number, $Ri = N^2/S^2$, may indicate conditions favourable for baroclinic instability (von Appen et al., 2016; Crews et al., 2018), as stratification is expected to suppress and shear to favour instability. We use the background fields of stratification, $N^2 = -(g/\rho_0)\partial\rho/\partial z$, and shear, $S^2 = (\partial\bar{u}/\partial z)^2$, instead of the thermal-wind shear, which are representative of the geostrophic scales. Overall, Ri was significantly smaller during winter than in summer, consistent with enhanced EKE and dominant BC in winter observed in the
mooring records. Median value of Ri was 2 to 20 times larger than in winter, depending on the mooring and the measurement level analysed. During winter, Ri was less than 50, 21 % of the time at W2 (30 % at W3), but only 5 % of the time during summer. Strong vertical shear, connected to the horizontal density gradient through the thermal wind relation, was the main factor reducing Ri.

The mean flow advecting eddies can also explain differences between measured energy conversion and observed EKE.
Eddies generated in regions of high instability can be transported downstream by the mean circulation – for instance, eddies observed in the eastern Eurasian Basin have been traced as far back upstream as the Yermak Plateau, the area north of Svalbard, and the slope between Svalbard and Franz Josef Land (Pnyushkov et al., 2018) – resulting in enhanced EKE away from the generation sites (von Appen et al., 2016; Wekerle et al., 2020; Fer et al., 2023). Furthermore, some EKE can originate from meandering of the current, hence elevated EKE levels do not necessarily indicate local eddy occurrence (Wekerle et al., 2020;
Crews et al., 2018).

In summary, several factors complicate the interpretation of instability diagnostics, conversion rate estimates, and local EKE: the necessary condition for barotropic instability identifies regions susceptible to instability, but may still not coincide with regions of high positive barotropic conversion; local conversion rates may not align with local EKE due to advection of eddies from upstream; and EKE levels themselves may not reflect the eddy formation or occurrence.

## 4.5 Limitations of using mooring observations for conversion rate estimations

As noted by earlier studies (e.g., Fer et al., 2020, 2023), conversion rates based on mooring observations are subject to significant uncertainty and therefore should be interpreted with caution. A primary limitation is that moorings provide fixed-point measurements, which may not capture the full spatial and temporal variability of energy conversion. Model results show that conversion rates are patchy and intermittent, as illustrated by differences between volume-averaged rates and estimates from
virtual moorings or cross-slope segments. Sufficient averaging in time and space is therefore essential for reliable conversion rate estimates.

Another limitation is that mooring arrays do not allow estimation of all terms in the conversion rate equations. This limitation is illustrated using the estimates from the simplified formulas at virtual mooring and segments versus the full equations.





In addition, a comparison of the four terms in the BT formula (Eq. 2) showed that the two divergent terms were more often

positive, while the rotational terms were occasionally negative. In a similar comparison for the two terms in the BC formula (Eq. 3), the signs often opposed. Hence, a negative value in one term estimated from moorings does not preclude a positive total conversion rate, or vice versa. Even under the assumption that point-based estimates from one term adequately represent regional conditions, uncertainties would still arise from methodological and instrumental limitations associated with the mooring data: for example a larger mooring spacing relative to the Rossby radius will underestimate the isopycnal slope and thus

BC (von Appen et al., 2016); and similarly, inadequate resolution of the lateral velocity shear will underestimate BT.

     Our model results suggest that much of the conversion occurs deeper and further offshore than our moorings (Fig. 6). Nonetheless, a considerable fraction of the conversion occurs over the slope and at the depth sampled by the moorings, as supported by both analyses of virtual moorings (Fig. 8) and by comparing full-depth cross-slope segment estimates with those at limited-depth segments with shorter lateral span. The virtual mooring results further demonstrate that the substantial spread

observed with the BC rates at various depths (Fig. 5) is to be expected. As such, mooring-based estimates are better regarded as order-of-magnitude approximations indicating the possible conversion for that time in that region.

## 5   Summary and Conclusions

Motivated by the potential role of mesoscale eddies in transporting AW and heat offshore from the boundary current, we investigated mesoscale variability and energy conversion rates on the continental slope north of Svalbard. Mooring observations

revealed large differences between autumn–winter and spring–summer both in EKE and energy conversion rates (Figs. 3, 4, 5). The relatively energetic autumn and winter periods coincided with the strongest boundary current and warmest AW (Fig. 2). The largest conversion rates were primarily baroclinic (Figs. 4, 5), indicating baroclinic instability of the boundary current. Our findings from year-long measurements provide observational support to previous limited observations from a cruise (Våge et al., 2016) and inferences from a model (Crews et al., 2018), which suggested baroclinic instability and associated warm-core

eddies carrying AW offshore.

     To put these findings obtained from limited mooring observations into perspective, and to evaluate whether the conversion calculations are representative for the larger slope area, we used a 500-m horizontal resolution ocean model. The model results revealed that the mooring arrays were placed in energetic zones along the continental slope where considerable energy conversion can occur (Fig. 6). They also revealed that substantial conversion occurs also deeper and offshore of the mooring

arrays.

     Although mooring data indicated that baroclinic conversion exceeded barotropic conversion, these estimates are subject to uncertainties, for example due to limited sampling in vertical and cross-slope extents as well as the assumption of along-slope homogeneity. Model results provided important context, showing that both barotropic and baroclinic conversion can occur at comparable magnitudes, with substantial variability in time and space, and that considerable energy conversion may take place

further offshore than the mooring arrays. These findings highlight the need for broad spatial sampling and adequate averaging to obtain representative estimates of energy conversion on the slope. Furthermore, conversion rates were only weakly connected



to local EKE variability in both mooring observations and model data, suggesting that advection of eddies from upstream unstable regions by the mean circulation may be important.

Our inferences from the mesoscale-band variability and the energetics help explain the missing along-stream heat loss from the boundary current in winter and spring that was reported in Koenig et al. (2022b) using the same moorings. Enhanced mesoscale activity in winter, and to a lesser extent in spring, facilitated mesoscale eddy stirring and lateral heat exchange, which become particularly important in periods when the AW has subducted and isolated from the atmosphere by sea ice cover.

*Data availability.* The Nansen Legacy mooring data (W1–W3, E1–E3, Fer et al., 2022) are available at the Norwegian Marine Data Centre:
doi.org/10.21335/NMDC-1852831792. ROMS model data is stored at IMR data servers and can be made available upon request. CARRA (Schyberg et al., 2020) data are available from doi.org/10.24381/cds.713858f6.

*Author contributions.* KK analysed the mooring data and performed calculations on the model output with support from IF and TMB. JA and LF set up and ran the ocean model. IF developed the research idea. IF and TMB provided guidance through the project. KK prepared the original draft with advice and support from IF. All authors discussed the results and finalized the paper.

*Competing interests.* Ilker Fer is a member of the editorial board of Ocean Science.

*Acknowledgements.* This work was funded by the Research Council of Norway through the project The Nansen Legacy (RCN # 276730). We thank the captains, officers, and crew on the research vessel *Kronprins Haakon* and our scientific and technical colleagues on the cruises for good cooperation during the deployment and recovery of the moorings. We are also grateful to two colleagues for reading the manuscript and providing valuable feedback.





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
