# Peer review of "Eddy kinetic energy and baroclinic and barotropic energy conversion rates along the Atlantic Water boundary current north of Svalbard"

_EGUsphere, 2025_

## Referee Comment (RC1)

The manuscript investigates the mechanisms of baroclinic and barotropic instabilities in the Atlantic Water boundary current north of Svalbard and their relationship to the variability of eddy kinetic energy (EKE). Using year-long mooring observations combined with output from an eddy-resolving ocean model, the authors examine the seasonal evolution of EKE and associated energy conversion rates. While the topic is scientifically relevant and the dataset is valuable, the manuscript requires major revision to improve clarity, strengthen the connection between observations and model results, and more convincingly support the conclusions drawn.

**General comments:**

- 1) The presented analysis of the mechanisms of baroclinic and barotropic energy conversion is thorough and certainly of interest to the oceanographic community. However, I find the study somewhat incomplete, as it does not provide sufficient insight into the consequences of these instabilities, which are potentially manifested as sub- and mesoscale eddies. The authors themselves point out that the link between the barotropic and baroclinic conversion rates and the eddy kinetic energy is rather weak, raising questions about the actual importance of these processes, at least within the specific region where the mooring array was deployed. Furthermore, the model simulations employed in this study do not appear to substantially advance our understanding of how the diagnosed energy conversions translate into eddy generation or eddy-induced heat transport. Given this weak connection with eddy processes, I would appreciate it if the authors could further elaborate on why understanding barotropic and baroclinic instabilities remains important in this context and what broader implications their results may have for the local or regional ocean dynamics.
- 2) Even if the observational dataset and its processing have been previously described and published elsewhere, I would recommend that the authors include a concise but sufficient description of the data and methodology within this manuscript to ensure it is self-contained and understandable to the reader. In particular, information about the vertical coverage and resolution of the instruments should be provided. For example, it is not clear how the distributions of current speed, temperature, and density in Fig. 2 were obtained, given that a substantial portion of the section is not covered by ADCP or T,S measurements. Please clarify how these data gaps were handled and indicate the potential uncertainties this may introduce into the analysis. Including these details would significantly improve the transparency of the study.
- 3) Taking into account that a portion of the observed EKE likely results from the advection of mesoscale eddies past the moorings, as indicated in the text, I am uncertain whether the selected cut-off period of 35 hours is appropriate for capturing the relevant variability. Assuming that the lateral scale of the advected eddies is approximately twice the local baroclinic Rossby radius of deformation (~7 km) and that the typical boundary current velocity is around 15 cm s-1, the corresponding advection time scale would be about 24 hours. Therefore, the applied filtering threshold might artificially attenuate the contribution of advected mesoscale features, leading to an underestimation of EKE and, consequently, of the lateral eddy-induced heat transport. It would be helpful if the authors could assess the sensitivity of their results to the choice of the band-pass filter parameters. Given that model output is available, it should be possible to

estimate the typical advection timescale directly from the simulated flow field and thereby verify whether the adopted cut-off period adequately captures the relevant eddy variability.

4) Before undertaking an in-depth analysis of the barotropic (BT) and baroclinic (BC) energy conversion rates, it is essential to demonstrate that the model reliably reproduces the mean structure of the boundary current and its baroclinic structure in agreement with observations. I recommend that the authors include direct model—observation comparisons and provide quantitative statistics of the mismatch. Useful additions would be: cross-slope sections of mean velocity and temperature/density from the model as well as vertical profiles of mean velocity at mooring locations. These comparisons should be shown for relevant seasonal subsets used in the conversion analysis. Demonstrating that the model adequately captures the observed mean flow and stratification is a prerequisite to attributing conversion rates and interpreting mesoscale energetics with confidence.

**Specific questions and suggestions:**

- **Table 1.** What do the numbers following the dates represent?
- **L.84.** Please make sure that your model indeed uses a terrain-following coordinate system, as stated, rather than a stretched one (s-coordinate).
- **Section 2.1.** It is unclear why the analysis is limited to the 300–700 m depth range, given that the cross-section includes data for the entire water column up to the surface. Is this restriction related to the actual (non-interpolated) data coverage? This limitation excludes the most energetic part of the boundary current near its core, where the dynamics are likely most intense. Moreover, even though the shallowest moorings (W1 and E1) are separated by a distance exceeding the baroclinic radius of deformation, incorporating data from these sites would substantially enhance the analysis and provide valuable insight into the asymmetry of EKE and baroclinic/barotropic energy conversion.
- **Figure 2.** Please label the mooring positions directly on the section plots.
- **L.185.** Where are the BT conversion rates in the upper water column shown?
- **Section 3.3.** The temporal and vertical averaging applied to the model output appears inconsistent with that used for the mooring observations, making direct comparison difficult. Could this discrepancy explain the significant mismatch between EKE and conversion rates derived from observations and from the model?
- **Section 4.3.** This part of the discussion appears insufficiently supported by the results presented in the manuscript and relies largely on reiteration of findings previously reported by Koenig et al. (2022). It is unclear what new insights regarding lateral eddy-induced transport are provided by this study. Moreover, according to the authors' own conclusions, the substantial BC/BT conversion rates are not directly linked to EKE, and thus are not clearly related to mesoscale eddy activity—an essential mechanism for Atlantic Water ventilation. In its current

form, this section seems exceptional and disconnected from the presented results; therefore, I suggest removing it altogether.

**Section 4.4.** For the negative BT conversion rates, which indicate energy transfer from eddies back to the mean current, was such re-energization reproduced by the model? Do you observe a statistically significant intensification of the boundary current in the regions where negative BT values occur? Demonstrating this would help make the discussion less abstract and more focused on the underlying physical processes.

---

## Author Comment (AC1)

**Response to Andrey Pnyushkov's comments**

The manuscript investigates the mechanisms of baroclinic and barotropic instabilities in the Atlantic Water boundary current north of Svalbard and their relationship to the variability of eddy kinetic energy (EKE). Using year-long mooring observations combined with output from an eddy-resolving ocean model, the authors examine the seasonal evolution of EKE and associated energy conversion rates. While the topic is scientifically relevant and the dataset is valuable, the manuscript requires major revision to improve clarity, strengthen the connection between observations and model results, and more convincingly support the conclusions drawn.

Thank you for taking the time to provide such a thorough review of our manuscript. We greatly appreciate your clear and concrete suggestion for improving clarity and strengthening the connection between the model and the observational results. Below, we outline how we have implemented many of your recommendations and how we plan to address the remaining points during the revision.

General comments:

1) The presented analysis of the mechanisms of baroclinic and barotropic energy conversion is thorough and certainly of interest to the oceanographic community. However, I find the study somewhat incomplete, as it does not provide sufficient insight into the consequences of these instabilities, which are potentially manifested as sub- and mesoscale eddies. The authors themselves point out that the link between the barotropic and baroclinic conversion rates and the eddy kinetic energy is rather weak, raising questions about the actual importance of these processes, at least within the specific region where the mooring array was deployed. Furthermore, the model simulations employed in this study do not appear to substantially advance our understanding of how the diagnosed energy conversions translate into eddy generation or eddy-induced heat transport. Given this weak connection with eddy processes, I would appreciate it if the authors could further elaborate on why understanding barotropic and baroclinic instabilities remains important in this context and what broader implications their results may have for the local or regional ocean dynamics.

We are pleased that you found our analysis thorough and of interest to the community, and we thank you for raising important points that will help us strengthen the manuscript.

In our observations, the direct link between the conversion rates and EKE variability is indeed weak, except for agreement on a seasonal time scale. The model results provide some insight into this: point-based estimates of conversion rates must be interpreted with caution, as they are not directly representative of larger volumes. However, they can serve as order-of-magnitude indicators.

In the model, we do see some relation between EKE and conversion rates when averaged in time and along-isobath. This is referred to in the manuscript for example when highlighting peaks in the conversion rates and in the EKE in month 14, although this relation was weaker

than anticipated. We do not interpret this as evidence that the processes are unimportant, but rather as an indication that there are open questions regarding how best to analyse and quantify their influence.

We also recognise that the current analysis of the model data does not fully demonstrate how the energy conversions translate into eddy generation or heat transport induced by eddies. Our main motivation for using model data was to provide context for the observation-based estimates and to assess how representative the conversion rates were. However, we agree that the model output could be used more extensively to investigate how energy conversion causes EKE variability and heat exchange induced by eddy activity. We are extending our calculations to include eddy-band temperature fluxes for the 1400-m isobath analysis of the model output (Fig. 7) which can then be directly compared to EKE and the conversion rates, and we will expand the discussion accordingly.

2) Even if the observational dataset and its processing have been previously described and published elsewhere, I would recommend that the authors include a concise but sufficient description of the data and methodology within this manuscript to ensure it is self-contained and understandable to the reader. In particular, information about the vertical coverage and resolution of the instruments should be provided. For example, it is not clear how the distributions of current speed, temperature, and density in Fig. 2 were obtained, given that a substantial portion of the section is not covered by ADCP or T,S measurements. Please clarify how these data gaps were handled and indicate the potential uncertainties this may introduce into the analysis. Including these details would significantly improve the transparency of the study.

We have added a clarification at the end of Section 2.1 on how the sections of mean current speed, temperature, and density in Fig. 2 were obtained.

To increase transparency, Fig. 2 has also been revised. We now show the vertical coverage of the ADCP for both seasons. The 90 % shading was adjusted to 80 % because one mooring was several hundred metres with 89 % coverage in one season. The 50 % shading was reduced to 20 % to avoid suggesting that no data were available in the upper water column. Additional temperature loggers (visible in Fig. 1 in Koenig et al., 2022) were used to obtain the mean temperature section, but they are not indicated here because those levels recorded temperature only and were not included in the analysis.

Regarding the data gaps and related uncertainties, we used only levels where measurement coverage in time and across a pair of moorings was sufficient and near instrument target depths (for BC). A more detailed explanation is provided under the point about Section 2.1 below, and we will add a brief clarification in the revised manuscript.

3) Taking into account that a portion of the observed EKE likely results from the advection of mesoscale eddies past the moorings, as indicated in the text, I am uncertain whether the

selected cut-off period of 35 hours is appropriate for capturing the relevant variability. Assuming that the lateral scale of the advected eddies is approximately twice the local baroclinic Rossby radius of deformation (~7 km) and that the typical boundary current velocity is around 15 cm s⁻¹, the corresponding advection time scale would be about 24 hours. Therefore, the applied filtering threshold might artificially attenuate the contribution of advected mesoscale features, leading to an underestimation of EKE and, consequently, of the lateral eddy-induced heat transport. It would be helpful if the authors could assess the sensitivity of their results to the choice of the band-pass filter parameters. Given that model output is available, it should be possible to estimate the typical advection timescale directly from the simulated flow field and thereby verify whether the adopted cut-off period adequately captures the relevant eddy variability.

We explored the effect of decreasing the high-frequency cut-off period. To avoid contribution from the diurnal tidal band, particularly the energetic K1 constituents, and ideally also the O1 constituent (25.8 h), we applied a sharper FIR Kaiser filter instead of the original Butterworth filter. This allowed us to reduce the cut-off to 27 h without including significant diurnal energy. Advection time scales for different current speeds can be estimated in the range of 24 h to 40 h, so a 27 h cut-off captures much of this variability.

We will include a figure in the appendix comparing results from the pass-band between 27 h and 14 d with the original band of 35 h and 14 d, for EKE, BT, and BC. The differences are small. As an example, November–December mean EKE at 400 m depth increases from 52 to 56 cm2 s-2 after changing the cut-off from 35 h to 27 h.

4) Before undertaking an in-depth analysis of the barotropic (BT) and baroclinic (BC) energy conversion rates, it is essential to demonstrate that the model reliably reproduces the mean structure of the boundary current and its baroclinic structure in agreement with observations. I recommend that the authors include direct model–observation comparisons and provide quantitative statistics of the mismatch. Useful additions would be: cross-slope sections of mean velocity and temperature/density from the model as well as vertical profiles of mean velocity at mooring locations. These comparisons should be shown for relevant seasonal subsets used in the conversion analysis. Demonstrating that the model adequately captures the observed mean flow and stratification is a prerequisite to attributing conversion rates and interpreting mesoscale energetics with confidence.

Agreed. We will add figures of model–observation comparisons showing average velocity and hydrography at cross-slope sections and vertical profiles for the relevant seasons.

The model setup was originally used in Frank et al., 2025 (DOI: 10.5194/os-21-2419-2025) where it was validated with data from Isfjorden in Svalbard and from the West Spitsbergen Shelf. These data included three autumn cross-slope sections across the shelf break along 78N and two years of mooring data at hourly resolution at the mouth of Isfjorden. The simulations showed temperature biases (too low at the surface, too high at depth over the slope) and salinity biases (too low in the surface layer over the shelf and in the Atlantic Water over the slope). These biases largely compensated, resulting in good agreement in the density structure.

The density time series also had good agreement between mooring observations and the model. The inflow pattern into Isfjorden was well represented, though inflow speeds were slightly over-estimated. In the revised manuscript, we cross-reference Frank et al. and provide a short summary of their validation results.

Specific questions and suggestions:

Table 1. What do the numbers following the dates represent?

They are the times of day. Thank you for pointing out that it was unclear. We have changed them to the HH:MM format and also added "(UTC)" in the table caption as suggested by Reviewer 2.

L.84. Please make sure that your model indeed uses a terrain-following coordinate system, as stated, rather than a stretched one (s-coordinate).

The model was set up with stretched terrain-following s-coordinates, using stretching parameters theta_b = 0.1, theta_s = 8, and T_cline = 20. Vertical transformation equation 2 (Vtransform = 2) and stretching function 2 (Vstretching = 2) were used (see https://www.myroms.org/wiki/Vertical_S-coordinate). We clarified this in Section 2.2 by specifying "… 35 vertical stretched bathymetry-following levels".

Section 2.1. It is unclear why the analysis is limited to the 300–700 m depth range, given that the cross-section includes data for the entire water column up to the surface. Is this restriction related to the actual (non-interpolated) data coverage? This limitation excludes the most energetic part of the boundary current near its core, where the dynamics are likely most intense. Moreover, even though the shallowest moorings (W1 and E1) are separated by a distance exceeding the baroclinic radius of deformation, incorporating data from these sites would substantially enhance the analysis and provide valuable insight into the asymmetry of EKE and baroclinic/barotropic energy conversion.

The 300–700 m depth range was the best option for this analysis, mainly due to data constraints, especially for calculating lateral gradients. Temporal averaging required consistent coverage in time, and lateral gradients in BT and BC conversion rates required additionally good coverage at both moorings simultaneously. Limited ADCP coverage at the deepest moorings (W3 and E3) therefore restricted the depth range for BT and BC estimates.

We agree that including data from the shallower moorings can add insight, even though

accurate conversion rate estimates are not possible there. We will add a figure with EKE, eddy momentum fluxes, and eddy density fluxes from W0, W1, and E1 (excluding eddy density flux at W1, where the temperature and conductivity sensors were lost). Furthermore, as lateral gradients are not required for EKE and eddy fluxes, we can also show them for a larger depth range at the deeper moorings. Although eddy fluxes alone do not directly quantify energy conversion, they provide additional context on the vertical and cross-slope structure of the energetics.

Figure 2. Please label the mooring positions directly on the section plots.

We have added labels ("W1", "W2", etc.) above the black triangles in Fig. 2.

L.185. Where are the BT conversion rates in the upper water column shown?

The wording in this sentence was unclear, and we have revised it. What we intended was "higher in the water column," i.e., closer to 300 m. This curve is not shown explicitly but is included within the standard-error shading in Fig. 4. We present BT with shading to reduce the number of curves in the plot.

Section 3.3. The temporal and vertical averaging applied to the model output appears inconsistent with that used for the mooring observations, making direct comparison difficult. Could this discrepancy explain the significant mismatch between EKE and conversion rates derived from observations and from the model?

Volume-averaged EKE and conversion rates from the model output were obtained by averaging over large volumes and in regions of the slope where the conversion rates were substantial, typically deeper and further offshore than the moorings. This is by design, and provides a "ground truth" of sufficiently averaged conversion rates representative for the region, to compare with the undersampled mooring coverage.

However, EKE and conversion rates at the virtual moorings are calculated exactly as those from the mooring observations. In Fig. 8, EKE and conversion rates are shown for in situ moorings and virtual moorings, based on data at the same depths, using only the terms in the simplified Equations 4 and 5 (BT and BC), averaged over 300 m to 700 m (EKE and BT), and monthly averaged. This shows that there is both a discrepancy between observations and the model (that we partly attribute to different time periods covered and perhaps capturing different dynamics) and between the virtual moorings and the average along the 1400 m isobath (due to using all terms and averaging over a larger area).

Section 4.3. This part of the discussion appears insufficiently supported by the results presented

in the manuscript and relies largely on reiteration of findings previously reported by Koenig et al. (2022). It is unclear what new insights regarding lateral eddy-induced transport are provided by this study. Moreover, according to the authors' own conclusions, the substantial BC/BT conversion rates are not directly linked to EKE, and thus are not clearly related to mesoscale eddy activity—an essential mechanism for Atlantic Water ventilation. In its current form, this section seems exceptional and disconnected from the presented results; therefore, I suggest removing it altogether.

We acknowledge your concern regarding the limited link between our estimated conversion rates and EKE. While we do not find a direct causal relationship, there is agreement on a seasonal time scale. Offshore eddy temperature fluxes and flux divergence from the western array are larger in autumn and winter than in spring and summer. This was noted briefly in Section 3.1 but not discussed further in Section 4.3.

Although temperature fluxes estimated from a few points must be interpreted with caution, as rotational and divergent terms cannot be separated, they suggest that lateral heat fluxes are enhanced during the same seasons when EKE and conversion rates are elevated. Our results complement the earlier estimations of along-path heat loss and atmospheric heat loss from the boundary current by Koenig et al., 2022 and provide further insight into the seasonality of heat-loss mechanisms.

To address your concerns, we will strengthen Section 4.3 by expanding the discussion of seasonal lateral heat loss due to mesoscale eddy activity and by analysing the relationship between EKE, conversion rates, and lateral heat exchange in the model.

Section 4.4. For the negative BT conversion rates, which indicate energy transfer from eddies back to the mean current, was such re-energization reproduced by the model? Do you observe a statistically significant intensification of the boundary current in the regions where negative BT values occur? Demonstrating this would help make the discussion less abstract and more focused on the underlying physical processes.

We appreciate this question, as negative conversion rates and their interpretation are, to our knowledge, rarely discussed in the literature. In the expanded analysis of the model results, we will examine whether regions with sustained negative BT conversion rates are associated with intensification of the boundary current.

We note, however, that given the weak connection between conversion rates and EKE variability, it may be difficult to establish a strong link between negative BT and boundary current intensification.